# EQUAL EXPERIENCE IN RECOMMENDER SYSTEMS

## ABSTRACT

We explore the *fairness* issue that arises in recommender systems. Biased data due to inherent stereotypes of particular groups (e.g., male students' average rating on mathematics is often higher than that on humanities, and vice versa for females) may yield a limited scope of suggested items to a certain group of users. Our main contribution lies in the introduction of a novel fairness notion (that we call *equal experience*), which can serve to regulate such unfairness in the presence of biased data. The notion captures the degree of the equal experience of item recommendations across distinct groups. We propose an optimization framework that incorporates the fairness notion as a regularization term, as well as introduce computationally-efficient algorithms that solve the optimization. Experiments on synthetic and benchmark real datasets demonstrate that the proposed framework can indeed mitigate such unfairness while exhibiting a minor degradation of recommendation accuracy.

## 1 INTRODUCTION

Recommender systems are everywhere, playing a crucial role to support decision making and to decide what we experience in our daily life. One recent challenge concerning *fairness* arises when the systems are built upon biased historical data. Biased data due to polarized preferences of particular groups for certain items may often yield limited recommendation service. For instance, if female students exhibit high ratings on literature subjects and less interest in math and science relative to males, the subject recommender system trained based on such data may provide a narrow scope of recommended subjects to the female group, thereby yielding *unequal* experience. This unequal experience across groups may result in amplifying the gender gap issue in science, technology, engineering, and mathematics (STEM) fields.

Among various works for fair recommender systems (Yao & Huang, 2017; Li et al., 2021; Kamishima & Akaho, 2017; Xiao et al., 2017; Beutel et al., 2019; Burke, 2017), one recent and most relevant work is Yao & Huang (2017). They focus on a scenario in which unfairness occurs mainly due to distinct recommendation accuracies across different groups. They propose novel fairness measures that quantify the degree of such unfairness via the difference between recommendation accuracies, and also develop an optimization framework that well trades the fairness measures against the average accuracy. However, it comes with a challenge in ensuring fairness w.r.t. the *unequal experience*. This is because similar accuracy performances between different groups do not guarantee a variety of recommendations to an underrepresented group with historical data bearing low preferences and/or scarce ratings for certain items. For instance, in the subject recommendation, the fairness notion may not serve properly, as long as female students exhibit low ratings (and/or lack of ratings) on math and science subjects due to societal/cultural influences (and/or sampling biases). Furthermore, if the recommended items are selected only according to the overall preference, the biased preference for a specific item group will further increase, and the exposure to the unpreferred item group will gradually decrease.

**Contribution:** In an effort to address the challenge, we introduce a new fairness notion that we call *equal experience*. At a high level, the notion represents how equally various items are suggested even for an underrepresented group preserving such biased historical data. Inspired by an information-theoretic notion *"mutual information"* (Cover, 1999) and its key property *"chain rule"*, we quantify our notion so as to control the level of *independence* between preference predictions and items for *any group* of users. Specifically, the notion encourages prediction $\widetilde{Y}$ (e.g., 1 if a user prefers an item; 0 otherwise) to be independent of the following two: (i) user group $Z_{\text{user}}$ (e.g., 0 for male; and 1 for

female); and (ii) item group $Z_{\text{item}}$ (e.g., 0 for mathematics; and 1 for literature). In other words, it promotes $\widetilde{Y} \perp (Z_{\text{user}}, Z_{\text{item}})$; which in turns ensures all of the following four types of independence that one can think of: $\widetilde{Y} \perp Z_{\text{item}}$, $\widetilde{Y} \perp Z_{\text{user}}$, $\widetilde{Y} \perp Z_{\text{item}}|Z_{\text{user}}$, and $\widetilde{Y} \perp Z_{\text{user}}|Z_{\text{item}}$. This is inspired by the fact that mutual information being zero is equivalent to the independence between associated random variables, as well as the chain rule:

$$
\begin{aligned}
I(\widetilde{Y}; Z_{\text{user}}, Z_{\text{item}}) &= I(\widetilde{Y}; Z_{\text{item}}) + I(\widetilde{Y}; Z_{\text{user}}|Z_{\text{item}}) \\
&= I(\widetilde{Y}; Z_{\text{user}}) + I(\widetilde{Y}; Z_{\text{item}}|Z_{\text{user}}).
\end{aligned}
\tag{1}
$$

See Section 3.1 for details. The higher independence, the more diverse recommendation services are offered for every group. We also develop an optimization framework that incorporates the quantified notion as a regularization term into a conventional optimization for collaborative filtering in recommender systems (e.g., the one based on matrix completion Koren (2008); Koren et al. (2009)). Here one noticeable feature of our framework is that the fairness performances w.r.t. the above *four* types of independence conditions can be gracefully controlled via a *single unified* regularization term. This is in stark contrast to prior works (Yao & Huang, 2017; Li et al., 2021; Kamishima & Akaho, 2017; Mehrotra et al., 2018), each of which promotes only one independence condition or two via two separate regularization terms. See below **Related works** for details. In order to enable an efficient implementation of the fairness constraint, we employ recent methodologies developed in the context of fair classifiers, such as the ones building upon kernel density estimation (Cho et al., 2020a), mutual information (Zhang et al., 2018; Kamishima et al., 2012; Cho et al., 2020b), or covariance (Zafar et al., 2017a;b). We also conduct extensive experiments both on synthetic and two benchmark real datasets: MovieLens 1M (Harper & Konstan, 2015) and Last FM 360K (Celma, 2010). As a result, we first identify two primary sources of biases that incur *unequal experience*: population imbalance and observation bias (Yao & Huang, 2017). In addition, we demonstrate that our fairness notion can help improve the fairness measure w.r.t. *equal experience* (to be defined in Section 3.1; see Definition 2) while exhibiting a small degradation of recommendation accuracy.

**Related works:** In addition to Yao & Huang (2017), numerous fairness notions and algorithms have been proposed for fair recommender systems (Xiao et al., 2017; Beutel et al., 2019; Singh & Joachims, 2018; Zehlike et al., 2017; Narasimhan et al., 2020; Biega et al., 2018; Li et al., 2021; Kamishima & Akaho, 2017; Mehrotra et al., 2018; Schnabel et al., 2016). Xiao et al. (2017) develop fairness notions that encourage similar recommendations for users within the same group. Beutel et al. (2019) consider similar metrics as that in Yao & Huang (2017) yet in the context of pairwise recommender systems wherein pairewise preferences are given as training data. Li et al. (2021) propose a fairness measure that quantifies the irrelevancy of preference predictions to user groups, like *demographic parity* in the fairness literature (Feldman et al., 2015; Zafar et al., 2017a;b). Specifically, they consider the independence condition between prediction $\widetilde{Y}$ and user group $Z_{\text{user}}$: $\widetilde{Y} \perp Z_{\text{user}}$. Actually this was also considered as another fairness measure in Yao & Huang (2017). Similarly, other works with a different direction consider the similar notion concerning the independence w.r.t. item group $Z_{\text{item}}$: $\widetilde{Y} \perp Z_{\text{item}}$ (Kamishima & Akaho, 2017; Singh & Joachims, 2018; Biega et al., 2018). Mehrotra et al. (2018) incorporate both measures to formulate a multi-objective optimization. In Section 2.2, we will elaborate on why the above prior fairness notions cannot fully address the challenge w.r.t. *unequal experience*.

There has been a proliferation of fairness notions in the context of fair classifiers: (i) group fairness (Feldman et al., 2015; Zafar et al., 2017b; Hardt et al., 2016; Woodworth et al., 2017); (ii) individual fairness (Dwork et al., 2012; Garg et al., 2018); (iii) causality-based fairness (Kusner et al., 2017; Nabi & Shpitser, 2018; Russell et al., 2017; Wu et al., 2019; Zhang & Bareinboim, 2018b;a). Among various prominent group fairness notions, *demographic parity* and *equalized odds* give an inspiration to our work in the process of applying the chain rule, reflected in equation 1. Concurrently, a multitude of fairness algorithms have been developed with the use of covariance (Zafar et al., 2017a;b), mutual information (Zhang et al., 2018; Kamishima et al., 2012; Cho et al., 2020b), kernel density estimation (Cho et al., 2020a) or Rényi correlation (Mary et al., 2019) to name a few. In this work, we also demonstrate that our proposed framework (to be presented in Section 3) embraces many of these approaches; See Remark 1 for details.

## 2 PROBLEM FORMULATION

As a key technique for operating recommender systems, we consider collaborative filtering which estimates user ratings on items. We first formulate an optimization problem for collaborative filtering building upon one prominent approach, *matrix completion*. We then introduce a couple of fairness measures proposed by recent prior works (Yao & Huang, 2017; Li et al., 2021; Kamishima & Akaho, 2017), and present an extended optimization framework that incorporates the fairness measures as regularization terms.

### 2.1 OPTIMIZATION BASED ON MATRIX COMPLETION

As a well-known approach for operating recommender systems, we consider matrix completion (Fazel, 2002; Koren et al., 2009; Candès & Recht, 2009). Let $M \in \mathbb{R}^{n \times m}$ be the ground-truth rating matrix where $n$ and $m$ denote the number of users and items respectively. Each entry, denoted by $M_{ij}$, can be of any type. It could be binary, five-star rating, or any real number. Denote by $\Omega$ the set of observed entries of $M$. For simplicity, we assume noiseless observation. Denote by $\widehat{M} \in \mathbb{R}^{n \times m}$ an estimate of the rating matrix.

Matrix completion can be done via the rank minimization that exploits the low-rank structure of the rating matrix. However, since the problem is NP-hard (Fazel, 2002), we consider a well-known relaxation approach that intends to minimize instead the squared error between $M$ and $\widehat{M}$ in the observed entries:

$$\min_{\widehat{M}} \sum_{(i,j) \in \Omega} (M_{ij} - \widehat{M}_{ij})^2. \tag{2}$$

There are two well-known approaches for solving the optimization in equation 2: (i) matrix factorization (Abadir & Magnus, 2005; Koren et al., 2009); and (ii) neural-net-based parameterization (Salakhutdinov et al., 2007; Sedhain et al., 2015; He et al., 2017). Matrix factorization assumes a certain structure on the rating matrix: $M = LR$ where $L \in \mathbb{R}^{n \times r}$ and $R \in \mathbb{R}^{r \times m}$. One natural way to search for optimal $L^*$ and $R^*$ is to apply gradient descent (Robbins & Monro, 1951) w.r.t. all of the $L_{ij}$'s and $R_{ij}$'s, although it does not ensure the convergence of the optimal point due to non-convexity. The second approach is to parameterize $\widehat{M}$ via neural networks such as restricted Boltzmann machine (Salakhutdinov et al., 2007) and autoencoder (Sedhain et al., 2015; Lee et al., 2018). For instance, one may employ an autoencoder-type neural network which outputs a completed matrix $\widehat{M}$ fed by the partially-observed version of $M$. For a user-based autoencoder (Sedhain et al., 2015), an observed *row* vector of $M$ is fed into the autoencoder, while an observed *column* vector serves as an input for an item-based autoencoder (Sedhain et al., 2015). In this work, we consider the two approaches in our experiments: matrix factorization with gradient descent; and autoencoder-based parameterization.

One common way to promote a *fair* recommender system is to incorporate a fairness measure, say $\mathcal{L}_{\mathsf{fair}}$ (which we will relate to an estimated matrix $\widehat{M}$), as a regularization term into the above base optimization in equation 2:

$$\min_{\widehat{M}} (1 - \lambda) \sum_{(i,j) \in \Omega} (M_{ij} - \widehat{M}_{ij})^2 + \lambda \cdot \mathcal{L}_{\mathsf{fair}} \tag{3}$$

where $\lambda \in [0, 1]$ denotes a normalized regularization factor that balances prediction accuracy against the fairness constraint. For the fairness-regularization term $\mathcal{L}_{\mathsf{fair}}$, several fairness measures have been introduced.

### 2.2 FAIRNESS MEASURES IN PRIOR WORKS (YAO & HUANG, 2017; KAMISHIMA & AKAHO, 2017; LI ET AL., 2021)

We list three of them, which are mostly relevant to our framework to be presented in Section 3. For illustrative purpose, we will explain them in a simple setting where there are two groups of users, say the male group $\mathcal{M}$ and the female group $\mathcal{F}$. The first is *value unfairness* proposed by Yao & Huang (2017). It quantifies the difference between prediction errors across the two groups of users over the

|  | $Z_{\text{item}} = 0$ | | | $Z_{\text{item}} = 1$ | | |
|---|---|---|---|---|---|---|
| | 1 | 1 | 1 | 0 | 0 | 0 |
| $Z_{\text{user}} = 0$ | 1 | 1 | 1 | 0 | 0 | 0 |
| | 1 | 1 | 1 | 0 | 0 | 0 |
| | 0 | 0 | 0 | 1 | 1 | 1 |
| $Z_{\text{user}} = 1$ | 0 | 0 | 0 | 1 | 1 | 1 |
| | 0 | 0 | 0 | 1 | 1 | 1 |

$\mathbf{1}\{\widehat{M}_{ij} \geq \tau\}$

$\widehat{Y}$: a generic random variable w.r.t. $\widehat{M}_{ij}$

$\widetilde{Y} := \mathbf{1}\{\widehat{Y} \geq \tau\}$

Figure 1: An example in which $\widetilde{Y} \perp Z_{\text{item}}$ but $\widetilde{Y} \not\perp Z_{\text{item}}|Z_{\text{user}}$. Here $\widetilde{Y} := \mathbf{1}\{\widehat{Y} \geq \tau\}$; $\widehat{Y}$ is a generic random variable w.r.t. estimated ratings $\widehat{M}_{ij}$'s; and $\tau$ indicates a certain threshold. The $(i, j)$ entry of an estimated preference matrix with 6 users (row) and 6 items (column) indicates $\mathbf{1}\{\widehat{M}_{ij} \geq \tau\}$.

entire items:

$$\text{VAL} := \frac{1}{m}\sum_{j=1}^{m}\left| \underbrace{\frac{1}{|\mathcal{M}_\Omega|}\sum_{(i,j)\in\Omega:i\in\mathcal{M}}(M_{ij} - \widehat{M}_{ij})}_{\text{prediction error w.r.t. } \mathcal{M}} - \underbrace{\frac{1}{|\mathcal{F}_\Omega|}\sum_{(i,j)\in\Omega:i\in\mathcal{F}}(M_{ij} - \widehat{M}_{ij})}_{\text{prediction error w.r.t. } \mathcal{F}} \right| \quad (4)$$

where $\mathcal{M}_\Omega$ and $\mathcal{F}_\Omega$ denote the male and female group w.r.t. observed entries, respectively. While the measure promotes fairness w.r.t. *prediction accuracy* across distinct groups, it may not ensure fairness w.r.t. the diversity of recommended items to users. To see this clearly, consider an extreme scenario in which the ground truth rating is very small $M_{ij^*} \approx 0$ for a certain item $j^*$ (say science subject) for all $i \in \mathcal{F}$. In this case, minimizing VAL may encourage $\widehat{M}_{ij^*} \approx 0$ for all $i \in \mathcal{F}$. This then incurs almost no recommendation of the science subject to the females, thus giving no opportunity to experience the subject. This motivates us to propose a new fairness measure (to be presented in Section 3.1) that helps mitigate such unfairness.

On the other hand, Kamishima & Akaho (2017) introduce another fairness measure, which bears a similar spirit to demographic parity in the fairness literature (Feldman et al., 2015; Zafar et al., 2017a;b). The measure, named *Calders and Verwer's discrimination score* (CVS), quantifies the level of irrelevancy between preference predictions and item groups. To describe it in detail, let us introduce some notations. Let $Z_{\text{item}}$ be a sensitive attribute w.r.t. item groups, e.g., $Z_{\text{item}} = 0$ (literature) and $Z_{\text{item}} = 1$ (science). Let $\widehat{Y}$ be a generic random variable w.r.t. estimated ratings $\widehat{M}_{ij}$'s. To capture the preference prediction, let us consider a simple binary preference setting in which $\widetilde{Y} := \mathbf{1}\{\widehat{Y} \geq \tau\}$ where $\tau$ indicates a certain threshold. Specializing the measure into the one like demographic parity, it can be quantified as:

$$\text{CVS} := |\mathbb{P}(\widetilde{Y} = 1|Z_{\text{item}} = 1) - \mathbb{P}(\widetilde{Y} = 1|Z_{\text{item}} = 0)|. \quad (5)$$

Minimizing the measure encourages the independence between $\widetilde{Y}$ and $Z_{\text{item}}$, thereby promoting the same rating statistics across different groups. However, it does not necessarily ensure the same statistics when we focus on a *certain* group of users. It guarantees the independence only in the *average* sense. To see this clearly, consider a simple scenario in which there are two groups of users, say female and male. Let $Z_{\text{user}}$ be another sensitive attribute w.r.t. user groups, e.g., $Z_{\text{user}} = 0$ (female) and $Z_{\text{user}} = 1$ (male). Fig. 1 illustrates a concrete example where $\widetilde{Y}$ is independent of $Z_{\text{item}}$. Notice that the number of 1's w.r.t. $Z_{\text{item}} = 0$ (over the entire user groups) is the same as that w.r.t. $Z_{\text{item}} = 1$. However, focusing on a certain user group, say $Z_{\text{user}} = 0$, $\widetilde{Y}$ is highly correlated with $Z_{\text{item}}$. Observe in the case $Z_{\text{user}} = 0$ that the number of 1's is 9 for $Z_{\text{item}} = 0$, while it reads 0 for $Z_{\text{item}} = 1$.

Li et al. (2021) consider a similar measure, named *user-oriented group fairness* (UGF), yet which targets the independence w.r.t. *user* groups. Similar to CVS, we can define it by replacing $Z_{\text{item}}$ with $Z_{\text{user}}$ in equation 5:

$$\text{UGF} := |\mathbb{P}(\widetilde{Y} = 1|Z_{\text{user}} = 1) - \mathbb{P}(\widetilde{Y} = 1|Z_{\text{user}} = 0)|. \quad (6)$$

However, by symmetry, the high correlation issue discussed via Fig. 1 still arises.

## 3 PROPOSED FRAMEWORK

We first propose new fairness measures that can regulate fairness w.r.t. the opportunity to experience inherently-low preference items, as well as address the high correlation issue discussed as above. We then develop an integrated optimization framework that unifies the fairness measures as a single regularization term. Finally we introduce concrete methodologies that can implement the proposed optimization.

### 3.1 NEW FAIRNESS MEASURES

The common limitation of the prior fairness measures (Yao & Huang, 2017; Kamishima & Akaho, 2017; Li et al., 2021) is that the independence between preference predictions and item groups may not be guaranteed for a *certain* group of users. This motivates us to consider the *conditional* independence as a new fairness notion, formally defined as below.

**Definition 1 (Equalized Recommendation)** *A recommender system is said to respect "equalized recommendation" if its prediction $\widetilde{Y}$ is independent of item's sensitive attribute $Z_{\text{item}}$ given user's sensitive attribute $Z_{\text{user}}$: $\widetilde{Y} \perp Z_{\text{item}}|Z_{\text{user}}$.*

Inspired by the quantification methods w.r.t. *equalized odds* in the fairness literature (Jiang et al., 2019; Donini et al., 2018; Hardt et al., 2016; Woodworth et al., 2017), we quantify the new notion via:

$$\text{DER} := \sum_{z_1 \in \mathcal{Z}_{\text{user}}} \sum_{z_2 \in \mathcal{Z}_{\text{item}}} \left| \mathbb{P}(\widetilde{Y} = 1|Z_{\text{user}} = z_1) - \mathbb{P}(\widetilde{Y} = 1|Z_{\text{item}} = z_2, Z_{\text{user}} = z_1) \right|, \quad (7)$$

for arbitrary alphabet sizes $|\mathcal{Z}_{\text{user}}|$ and $|\mathcal{Z}_{\text{item}}|$. Here DER stands for the difference w.r.t. two interested probabilities that arise in equalized recommendation, and this naming is similar to those in prior fairness metrics (Donini et al., 2018; Jiang et al., 2019). It captures the degree of violating equalized recommendation via the difference between the conditional probability and its marginal given $Z_{\text{user}}$. Notice that the minimum DER $= 0$ is achieved under "equalized recommendation". One may consider another measure which takes "max" operation instead of "$\sum$" in equation 7 or a different measure based on the *ratio* of the two associated probabilities. We focus on DER in equation 7 for tractability of an associated optimization problem that we will explain in Section 3.2.

The constraint of "equalized recommendation" encourages the same prediction statistics of items for *every* user group, thereby promoting the equal chances of experiencing a variety of items for all individuals. However, the notion comes with a limitation. The limitation comes from the fact that conditional independence does not necessarily imply independence (Cover, 1999):

$$\widetilde{Y} \perp Z_{\text{item}}|Z_{\text{user}} \implies \widetilde{Y} \perp Z_{\text{item}}. \quad (8)$$

Actually, the ultimate goal of a fair recommender system is to ensure all of the following four types of independence:

$$\widetilde{Y} \perp Z_{\text{item}}, \qquad \widetilde{Y} \perp Z_{\text{user}}|Z_{\text{item}}, \qquad \widetilde{Y} \perp Z_{\text{user}}, \qquad \widetilde{Y} \perp Z_{\text{item}}|Z_{\text{user}}. \quad (9)$$

One natural question that arises is then: What is a proper fairness notion which allows us to respect all of the above four conditions preferably in one shot? In an attempt to succinctly represent all of the four conditions, we invoke an information-theoretic notion, *mutual information* (Cover, 1999). One key property of mutual information, called the chain rule, gives an insight:

$$\begin{aligned} I(\widetilde{Y}; Z_{\text{user}}, Z_{\text{item}}) &= I(\widetilde{Y}; Z_{\text{item}}) + I(\widetilde{Y}; Z_{\text{user}}|Z_{\text{item}}) \\ &= I(\widetilde{Y}; Z_{\text{user}}) + I(\widetilde{Y}; Z_{\text{item}}|Z_{\text{user}}). \end{aligned} \quad (10)$$

From this, we can readily see that

$$\begin{aligned} I(\widetilde{Y}; Z_{\text{user}}, Z_{\text{item}}) = 0 \implies &I(\widetilde{Y}; Z_{\text{item}}) = 0, \ I(\widetilde{Y}; Z_{\text{user}}|Z_{\text{item}}) = 0, \\ &I(\widetilde{Y}; Z_{\text{user}}) = 0, \ I(\widetilde{Y}; Z_{\text{item}}|Z_{\text{user}}) = 0. \end{aligned} \quad (11)$$

This is due to the non-negativity property of mutual information. The key observation in equation 11 motivates us to propose a new fairness notion that we call *equal experience*.

**Definition 2 (Equal Experience)** *A recommender system is said to respect "equal experience" if its prediction $\widetilde{Y}$ is independent of both $Z_{\mathsf{item}}$ and $Z_{\mathsf{user}}$: $\widetilde{Y} \perp (Z_{\mathsf{item}}, Z_{\mathsf{user}})$.*

Similar to DER, we also quantify the notion as the difference between the conditional probability and its marginal:

$$\mathsf{DEE} := \sum_{z_1 \in \mathcal{Z}_{\mathsf{user}}} \sum_{z_2 \in \mathcal{Z}_{\mathsf{item}}} \left| \mathbb{P}(\widetilde{Y} = 1) - \mathbb{P}(\widetilde{Y} = 1 | Z_{\mathsf{item}} = z_2, Z_{\mathsf{user}} = z_1) \right|, \tag{12}$$

for arbitrary alphabet sizes $|\mathcal{Z}_{\mathsf{user}}|$ and $|\mathcal{Z}_{\mathsf{item}}|$. We also coin the similar naming: DEE (difference w.r.t. two interested probabilities that arise in equal experience).

### 3.2 Fairness-regularized optimization

Taking DEE as the fairness-regularization term $\mathcal{L}_{\mathsf{fair}}$ in the focused framework (equation 3), we get:

$$\min_{\widehat{M}} (1 - \lambda) \sum_{(i,j) \in \Omega} (M_{ij} - \widehat{M}_{ij})^2 + \lambda \cdot \mathsf{DEE} \tag{13}$$

where $\lambda \in [0, 1]$ denotes a normalized regularization factor. Here one challenge that arises in equation 13 is that expressing DEE in terms of an optimization variable $\widehat{M}$ is not that straightforward.

To overcome the challenge, we take the kernel density estimation (KDE) technique (Cho et al., 2020a) which enables faithful quantification of fairness-regularization terms. One key benefit of the KDE approach is that the computed measures based on KDE is differentiable w.r.t. model parameters, thus enjoying a family of gradient-based optimizers (Géron, 2019; Kingma & Ba, 2014b). Since the problem context where the KDE technique (Cho et al., 2020a) was introduced is different from ours, we describe below details on the technique, tailoring it to our framework.

**Implementation of DEE via the KDE technique (Cho et al., 2020a):** We first parameterize prediction output $\widehat{M}$ via matrix factorization or a neural network. Let $w$ be a collection of parameters w.r.t. $\widehat{M}$. It could be a collection of matrix entries of $L$ and $R$ when employing matrix factorization $\widehat{M} = LR$. Or it could be a collection of neural network parameters in the latter case.

The key idea of the KDE technique is to *approximate* the interested probability distributions via kernel density estimator defined below:

**Definition 3 (Kernel Density Estimator (KDE) (Davis et al., 2011))** *Let $(\hat{y}^{(1)}, \ldots, \hat{y}^{(s)})$ be i.i.d. examples drawn from a distribution with an unknown density $f$. Its KDE is defined as: $\widehat{f}(\hat{y}) := \frac{1}{sh} \sum_{i=1}^{s} f_k \left( \frac{\hat{y} - \hat{y}^{(i)}}{h} \right)$ where $f_k$ is a kernel function (e.g., Gaussian kernel function (Davis et al., 2011)) and $h > 0$ is a smoothing parameter called bandwidth.*

For DEE, the interested probability distributions are $\mathbb{P}(\widetilde{Y} = 1)$ and $\mathbb{P}(\widetilde{Y} = 1 | Z_{\mathsf{item}} = z_2, Z_{\mathsf{user}} = z_1)$. Let us first consider $\mathbb{P}(\widetilde{Y} = 1)$. Remember $\widetilde{Y} := \mathbf{1}\{\widehat{Y} \geq \tau\}$, so $\widehat{Y}$ should be taken into consideration initially. Using the KDE, we can estimate the probability density function of $\widehat{Y}$, say $f_{\widehat{Y}}(\hat{y})$:

$$\widehat{f}_{\widehat{Y}}(\hat{y}) = \frac{1}{nmh} \sum_{i=1}^{n} \sum_{j=1}^{m} f_k \left( \frac{\hat{y} - \widehat{M}_{ij}}{h} \right). \tag{14}$$

This together with $\widetilde{Y} := \mathbf{1}\{\widehat{Y} \geq \tau\}$ gives:

$$\widehat{\mathbb{P}}(\widetilde{Y} = 1) = \int_{\tau}^{\infty} \widehat{f}_{\widehat{Y}}(\hat{y}) d\hat{y} = \frac{1}{nm} \sum_{i=1}^{n} \sum_{j=1}^{m} F_k \left( \frac{\tau - \widehat{M}_{ij}}{h} \right)$$

where $F_k(\hat{y}) := \int_{\hat{y}}^{\infty} f_k(t) dt$. Since the approach relies upon a family of gradient-based optimizers, the gradients of $\widehat{\mathbb{P}}(\widetilde{Y} = 1)$ and $\widehat{\mathbb{P}}(\widetilde{Y} = 1 | Z_{\mathsf{item}} = z_2, Z_{\mathsf{user}} = z_1)$ need to be computed explicitly. Using the technique in Cho et al. (2020a) (Proposition 1 therein), we can readily approximate $\nabla_w \mathsf{DEE}$. See Appendix A.2 for details.

**Remark 1 (Other choices for a measure of "equal experience")** *Instead of* DEE*, one can resort to other measures based on prominent tools employed in the fairness literature: covariance (Zafar et al., 2017a;b), mutual information (Zhang et al., 2018; Kamishima et al., 2012; Cho et al., 2020b), Wasserstein distance (Jiang et al., 2020) and Rényi correlation (Mary et al., 2019). We leave a more detailed explanation in Appendix A.3.* ■

## 4 EXPERIMENTS

We conduct experiments both on synthetic and two benchmark real datasets: MovieLens 1M (Harper & Konstan, 2015) and Last FM 360K (Celma, 2010). We generate synthetic data so as to pose fairness issues. Algorithms are implemented in PyTorch (Paszke et al., 2019), and experiments are performed on a server with Titan RTX GPUs. All the simulation results (to be reported) are the ones averaged over five trials with distinct random seeds. In Appendix A.6, we also present the running times of our algorithm and baselines on the synthetic and real datasets.

### 4.1 SYNTHETIC DATASET

Here we highlight two types of bias: population imbalance and observation bias (Yao & Huang, 2017). For illustrative purpose, let us explain them in a simple subject-recommendation example where there are two user groups ($Z_{\text{user}} = 0$ for male and $Z_{\text{user}} = 1$ for female) and two item groups ($Z_{\text{item}} = 0$ for science and $Z_{\text{item}} = 1$ for literature). Population imbalance refers to the difference in the ground-truth preferences between two user groups, e.g., for the science subject, male students exhibit higher ratings relative to females. Observation bias is the one that occurs due to the stereotype formed by societal and cultural influences. To understand what it means, let us consider a scenario where a male student equally likes science and literature subjects. But due to the stereotype that male students prefer science to literature in general, there may be very sparse ratings from male students for literature. The system trained based on such data might incorrectly interpret as if male students dislike literature. Such data is said to have *observation bias*.

We generate synthetic data that bear the two biases in the context of binary ratings, i.e., $M_{ij} \in \{+1 \text{ (like)}, -1 \text{ (dislike)}\}$. We divide $n$ users into male and female groups each of $\frac{n}{2}$, say $\mathcal{M}$ and $\mathcal{F}$. Items are also divided into two groups of $\frac{m}{2}$, say male-preferred group $\mathcal{M}'$ and female-preferred group $\mathcal{F}'$. To account for population imbalance, we first generate the ground-truth rating matrix $M \in \mathbb{R}^{n \times m}$ using the following four probabilities: $\{p_{\mathcal{M}\mathcal{M}'}, p_{\mathcal{M}\mathcal{F}'}, p_{\mathcal{F}\mathcal{M}'}, p_{\mathcal{F}\mathcal{F}'}\}$ where $p_{\mathcal{M}\mathcal{M}'}$ indicates the probability that a male student likes a male-preferred subject (science). More precisely, for $i \in \mathcal{M}$ and $j \in \mathcal{M}'$,

$$M_{ij} = \begin{cases} +1, & \text{w.p. } p_{\mathcal{M}\mathcal{M}'}; \\ -1, & \text{w.p. } 1 - p_{\mathcal{M}\mathcal{M}'}. \end{cases} \tag{15}$$

Similarly the other probabilities are defined. To ensure the low-rank structure, say rank $r$, of the rating matrix, we generate $\frac{r}{2}$ basis rating vectors for male group as per the above preference probabilities, and similarly another set of $\frac{r}{2}$ basis rating vectors is generated for female group. Every male student picks one of the $\frac{r}{2}$ basis vectors w.r.t. the male group uniformly at random, and similarly for every female student. This then yields $\text{rank}(\mathbf{M}) = r$.

To control observation bias, we introduce another probability set: $\{q_{\mathcal{M}\mathcal{M}'}, q_{\mathcal{M}\mathcal{F}'}, q_{\mathcal{F}\mathcal{M}'}, q_{\mathcal{F}\mathcal{F}'}\}$ where $q_{\mathcal{M}\mathcal{M}'}$ denotes the probability that a male student's rating is observed for a male-preferred subject. More precisely, for $i \in \mathcal{M}$ and $j \in \mathcal{M}'$,

$$(i, j) \in \begin{cases} \Omega, & \text{w.p. } q_{\mathcal{M}\mathcal{M}'}; \\ \Omega^c, & \text{w.p. } 1 - q_{\mathcal{M}\mathcal{M}'}. \end{cases} \tag{16}$$

Similarly the other probabilities are defined. For simplicity, throughout all the synthetic data simulations, we assume a symmetric setting in which $p_{\mathcal{M}\mathcal{M}'} = p_{\mathcal{F}\mathcal{F}'}(= p_0)$, $p_{\mathcal{M}\mathcal{F}'} = p_{\mathcal{F}\mathcal{M}'}(= p_1)$, $q_{\mathcal{M}\mathcal{M}'} = q_{\mathcal{F}\mathcal{F}'}(= q_0)$, and $q_{\mathcal{M}\mathcal{F}'} = q_{\mathcal{F}\mathcal{M}'}(= q_1)$.

We consider a setting in which $(r, n, m) = (20, 600, 400)$. We leave a more detailed explanation of experiments on the synthetic dataset in Appendix A.4. First, we check whether the bias actually incurs unfair recommendations. For ease of understanding, we consider two scenarios: (i) population imbalance varies without observation bias; and (ii) observation bias varies without population

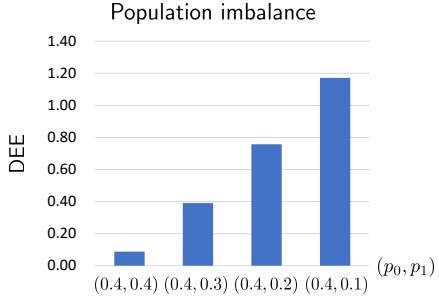
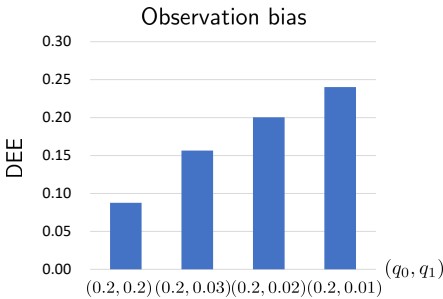

Figure 2: (Left) DEE as a function of $(p_0, p_1)$ which controls the degree of population imbalance while fixing $q_0 = q_1 = 0.2$; (Right) DEE as a function of $(q_0, q_1)$ w.r.t. observation bias while fixing $p_0 = p_1 = 0.4$. Here $p_0 = p_{\mathcal{MM'}} = p_{\mathcal{FF'}}$, $p_1 = p_{\mathcal{MF'}} = p_{\mathcal{FM'}}$, $q_0 = q_{\mathcal{MM'}} = q_{\mathcal{FF'}}$ and $q_1 = q_{\mathcal{MF'}} = q_{\mathcal{FM'}}$.

Table 1: Prediction error (RMSE) and fairness performances on the synthetic dataset preserving observation bias $(q_0, q_1) = (0.2, 0.01)$ while exhibiting no population imbalance $(p_0, p_1) = (0.4, 0.4)$. The boldface indicates the best result and the underline denotes the second best.

| Measure | RMSE | DEE | VAL | UGF | CVS |
|---|---|---|---|---|---|
| Unfair | $0.8889 \pm 0.0111$ | $0.1201 \pm 0.0405$ | $0.4646 \pm 0.0108$ | $0.0120 \pm 0.0050$ | $0.0118 \pm 0.0047$ |
| Ours (DEE) | $0.9020 \pm 0.0081$ | $\mathbf{0.0025 \pm 0.0006}$ | $0.4609 \pm 0.0076$ | $\underline{0.0006 \pm 0.0004}$ | $\underline{0.0002 \pm 0.0001}$ |
| Ours (DER) | $0.8887 \pm 0.0042$ | $\underline{0.0401 \pm 0.0056}$ | $0.4540 \pm 0.0110$ | $0.0201 \pm 0.0028$ | $\underline{0.0002 \pm 0.0002}$ |
| VAL-based | $0.8837 \pm 0.0045$ | $0.1099 \pm 0.0045$ | $\mathbf{0.0003 \pm 6.48e\text{-}6}$ | $0.0090 \pm 0.0004$ | $0.0088 \pm 0.0015$ |
| UGF-based | $0.8961 \pm 0.0067$ | $0.0144 \pm 0.0144$ | $0.4709 \pm 0.0182$ | $\mathbf{0.0004 \pm 0.0003}$ | $0.0217 \pm 0.0027$ |
| CVS-based | $0.9003 \pm 0.0061$ | $0.1390 \pm 0.0413$ | $0.4722 \pm 0.0055$ | $0.0206 \pm 0.0022$ | $\mathbf{0.0002 \pm 0.0001}$ |

imbalance. Fig. 2 (Left) presents the 1st scenario, demonstrating that the fairness performance measured in DEE decreases with an increase in population imbalance, controlled by $|p_0 - p_1|$. Fig. 2 (Right) considers the 2nd scenario. We see the same trend yet now w.r.t. the variation of observation bias.

Table 1 presents the prediction error (RMSE) and fairness performances on the synthetic dataset having observation bias $(q_0, q_1) = (0.2, 0.01)$ yet without population imbalance $(p_0, p_1) = (0.4, 0.4)$. We consider four fairness measures: (i) DEE (in equation 12); (ii) DER (in equation 7); (iii) VAL (in equation 4); (iv) UGF (in equation 6); (v) CVS (in equation 5). We also compare our algorithm with four baselines: (i) unfair (no fairness constraint); (ii) VAL-based algorithm (Yao & Huang, 2017); (iii) UGF-based algorithm (Li et al., 2021); (iv) CVS-based algorithm (Kamishima & Akaho, 2017). Each baseline, say VAL-based algorithm, achieves the best fairness performance only for VAL, while it does not work well under the other fairness measures. On the other hand, our algorithm offers great fairness performances for all the measures, except for VAL, which our framework does not target.

## 4.2 REAL DATASETS

We consider two benchmark datasets: MovieLens 1M (Harper & Konstan, 2015), and Last FM 360K (Celma, 2010). We run experiments employing both matrix factorization (MF) based and autoencoder (AE) based techniques. In Appendix A.5, we leave a more detailed explanation of experiments on the real datasets and the results of AE-based technique. The results of real data experiments are listed in Tables 2 and 3. As in the synthetic data setting, we can make two relevant observations. All baseline algorithms fail to respect our metric (DEE) while meeting their own. We also see that our algorithm exhibits significant performances for all the fairness measures except for VAL which does not have close relationship with the equal experience that we aim at.

Table 2: Prediction error (RMSE) and fairness performances of the matrix factorization based algorithm on *MovieLens 1M dataset*. The boldface indicates the best result and the underline denotes the second best. Our algorithm offers great fairness performances for all the measures, except for VAL, which our framework does not target.

| Measure | RMSE | DEE | VAL | UGF | CVS |
|---------|------|-----|-----|-----|-----|
| Unfair | $0.8541 \pm 0.0033$ | $0.2447 \pm 0.0134$ | $0.3227 \pm 0.0031$ | $0.0058 \pm 0.0042$ | $0.1291 \pm 0.0079$ |
| Ours (DEE) | $0.8641 \pm 0.0047$ | $\mathbf{0.0014 \pm 0.0008}$ | $0.2941 \pm 0.0024$ | $\underline{0.0018 \pm 0.0016}$ | $\underline{0.0007 \pm 0.0004}$ |
| Ours (DER) | $0.8526 \pm 0.0029$ | $0.0114 \pm 0.0041$ | $0.3332 \pm 0.0050$ | $\underline{0.0055 \pm 0.0022}$ | $0.0014 \pm 0.0001$ |
| VAL-based | $0.8529 \pm 0.0011$ | $0.3659 \pm 0.0033$ | $\mathbf{0.0942 \pm 0.0016}$ | $0.0261 \pm 0.0020$ | $0.1388 \pm 0.0030$ |
| UGF-based | $0.8550 \pm 0.0015$ | $0.2492 \pm 0.0100$ | $0.3285 \pm 0.0051$ | $\mathbf{0.0001 \pm 0.0001}$ | $0.1355 \pm 0.0038$ |
| CVS-based | $0.8549 \pm 0.0018$ | $0.0721 \pm 0.0069$ | $0.3319 \pm 0.0046$ | $0.0065 \pm 0.0042$ | $\mathbf{0.0002 \pm 3.45e\text{-}5}$ |

Table 3: Prediction error (RMSE) and fairness performances of the matrix factorization based algorithm on *Last FM 360K dataset*.

| Measure | RMSE | DEE | VAL | UGF | CVS |
|---------|------|-----|-----|-----|-----|
| Unfair | $0.6720 \pm 0.0024$ | $0.0840 \pm 0.0110$ | $0.2297 \pm 0.0020$ | $0.0404 \pm 0.0033$ | $0.0204 \pm 0.0104$ |
| Ours (DEE) | $0.6892 \pm 0.0040$ | $\mathbf{0.0082 \pm 0.0023}$ | $0.2777 \pm 0.0048$ | $\underline{0.0161 \pm 0.0009}$ | $\underline{0.0040 \pm 0.0011}$ |
| Ours (DER) | $0.6830 \pm 0.0033$ | $0.0711 \pm 0.0140$ | $0.2588 \pm 0.0024$ | $0.0356 \pm 0.0072$ | $0.0047 \pm 0.0007$ |
| VAL-based | $0.6802 \pm 0.0006$ | $0.1461 \pm 0.0216$ | $\mathbf{0.0016 \pm 2.03e\text{-}5}$ | $0.0234 \pm 0.0020$ | $0.0324 \pm 0.0202$ |
| UGF-based | $0.6705 \pm 0.0030$ | $0.0644 \pm 0.0313$ | $0.2366 \pm 0.0016$ | $\mathbf{0.0011 \pm 4.03e\text{-}5}$ | $0.3221 \pm 0.0157$ |
| CVS-based | $0.6758 \pm 0.0037$ | $0.0791 \pm 0.0136$ | $0.2448 \pm 0.0034$ | $0.0373 \pm 0.0063$ | $\mathbf{0.0012 \pm 0.0003}$ |

## 5 EXTENSION

In this section, we discuss the extension of our work: (i) introducing a new notion that bears a similar spirit to *equalized odds*, and (ii) applying our notion to the *fair ranking* context.

Our fairness notion *equal experience* aims at recommending a variety of items for all user groups. *Demographic parity* is similar to our notion in the sense of considering the irrelevancy of predictions to groups. On the other hand, *equalized odds* is the fairness notion that concerns equal error rates (e.g., true/negative positive rate) across user groups by employing the ground-truth label $Y$. Similar to *equalized odds*, our measure in recommender systems can readily be extended to a setting in which the ground-truth label is available to exploit. The key idea is to promote $\widetilde{Y} \perp (Z_{\text{user}}, Z_{\text{item}})|Y$. See Appendix A.7 for details.

Many end-to-end recommender systems offer a recommendation list via two processes: (i) candidate generation, and (ii) ranking. In this work, we focus on the first candidate generation which we built collaborative filtering for. But our proposed notion can also be applicable in generating an end ranked list. The idea behind the end-ranked list generation is to define $\widetilde{Y}$ as an indicator function which returns 1 when the item of interest belongs to, say top-k item set (0 otherwise). In this case, the same notion $\widetilde{Y} \perp (Z_{\text{user}}, Z_{\text{item}})$ serves a proper role. We leave detailed implementation of this notion under the fair ranking context for a future work.

## 6 CONCLUSION

We introduced a novel fairness notion, *equal experience*, capable of respecting the desired requirements for fair recommender systems: independence between preference predictions and user groups; conditional independence for a certain user group, and vice versa for item groups. The notion also seamlessly integrates into prior fairness algorithms. Extensive experiments revealed the existence of unfairness (or bias) w.r.t. *equal experience*, and our fair optimization framework successfully mitigates such unfairness with minimal prediction accuracy loss. Our future work of interest is four-folded: (i) merging our notion with unexamined algorithms relying upon Rényi correlation or Wasserstein distance; (ii) constructing a *robust* and fair recommender system in the presence of data poisoning; (iii) developing a blind fair recommender system without sensitive attributes; and (iv) extending our notion to fair ranking.

ETHICS STATEMENT

The proposed fairness notion and framework (which faithfully respects the notion) will enable a recommender system to offer an enough opportunity for users to experience various groups of items against stereotypes. Users are further influenced by the various experiences when making more essential choices such as choosing a job. Hence, it can give significant impacts upon users who can expand their options via diverse experiences. On the other hand, users provide sensitive information to a service provider of fair recommender systems, and there is a possibility that the user's sensitive information could be abused. Hence, one future work of great potential might be to develop a framework that ensures fairness even in the blind setting where sensitive information is not available.

REPRODUCIBILITY STATEMENT

To ensure the reproducibility of experiments, we provide a step-by-step guideline as to synthetic data construction in Section 4.1. We also provide a thorough description regarding two real datasets, MovieLens 1M (Harper & Konstan, 2015) and Last FM 360K (Celma, 2010), along with data preprocessing methods in Section 4.2. All the hyperparameter settings w.r.t. synthetic and real data experiments are listed in Appendix A.4 and A.5, respectively. We present the average running time of all algorithms together with specific computer configuration details in Appendix A.6. Lastly, the codes are available at `https://github.com/cjw2525/FairRec`.

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

## A  APPENDIX

### A.1  OUTLINE

In the appendix, we first provide further explanation for the implementation of KDE technique. We also present other choices for a measure of *equal experience* in further detail. Next, we provide a detailed explanation of experimental settings on both synthetic and real datasets: MovieLens 1M (Harper & Konstan, 2015) and Last FM 360K (Celma, 2010). We then present additional experimental results which are not included in the main paper due to space limitation, and as well as provide a complexity analysis of our approach. We also provide a detailed explanation for the extension of our measure to the fairness notion: $\widetilde{Y} \perp (Z_{\text{user}}, Z_{\text{item}})|Y$, and present experimental results for this extension.

### A.2  IMPLEMENTATION OF KDE TECHNIQUE

In order to approximate $\nabla_w \mathsf{DEE}$, we first calculate the gradient of $\widehat{\mathbb{P}}(\widetilde{Y} = 1)$ employing the technique in Cho et al. (2020a):

$$\nabla_w \widehat{\mathbb{P}}(\widetilde{Y} = 1) = \frac{1}{nmh} \sum_{i=1}^{n} \sum_{j=1}^{m} f_k \left( \frac{\tau - \widehat{M}_{ij}}{h} \right) \cdot \nabla_w \widehat{M}_{ij}. \tag{17}$$

We can apply the same procedures w.r.t. the second interested probability $\widehat{\mathbb{P}}(\widetilde{Y} = 1|Z_{\text{item}} = z_2, Z_{\text{user}} = z_1)$. Merging equation 17 and the counterpart w.r.t. the second probability, we can readily obtain:

$$\nabla_w \mathsf{DEE} \approx \sum_{z_1 \in \mathscr{Z}_{\text{user}}} \sum_{z_2 \in \mathscr{Z}_{\text{item}}} H'_\delta \left( \widehat{\mathbb{P}}(\widetilde{Y} = 1|Z_{\text{item}} = z_2, Z_{\text{user}} = z_1) - \widehat{\mathbb{P}}(\widetilde{Y} = 1) \right)$$
$$\times \nabla_w \left( \widehat{\mathbb{P}}(\widetilde{Y} = 1|Z_{\text{item}} = z_2, Z_{\text{user}} = z_1) - \widehat{\mathbb{P}}(\widetilde{Y} = 1) \right) \tag{18}$$

where $H_\delta(x)$ denotes the Huber loss (Huber, 1992) that takes $\frac{1}{2}x^2$ when $|x| \leq \delta$; otherwise $\delta(|x| - \frac{1}{2}\delta)$.

### A.3  OTHER CHOICES FOR A MEASURE OF *"equal experience"*

Instead of DEE in equation 12, one can resort to other measures based on prominent tools employed in the fairness literature: covariance (Zafar et al., 2017a;b), mutual information (Zhang et al., 2018; Kamishima et al., 2012; Cho et al., 2020b), Wasserstein distance (Jiang et al., 2020) and Rényi correlation (Mary et al., 2019). For instance, the covariance-based approach allows us to take $\mathcal{L}_{\text{fair}}$ as:

$$\mathcal{L}_{\text{fair}} = \mathbb{E}\left[ (\widehat{Y} - \mathbb{E}[\widehat{Y}])(Z - \mathbb{E}[Z]) \right] \qquad \text{where } Z := (Z_{\text{item}}, Z_{\text{user}}). \tag{19}$$

Here we use $\widehat{Y}$ instead of $\widetilde{Y}$, as $\widetilde{Y}$ incurs non-differentiability, hindering implementation. In the case of mutual information, one can take $\mathcal{L}_{\text{fair}}$ as:

$$\mathcal{L}_{\text{fair}} = I(\widehat{Y}; Z_{\text{item}}, Z_{\text{user}}) \geq I(\widetilde{Y}; Z_{\text{item}}, Z_{\text{user}}). \tag{20}$$

Again, for ease of implementation, we employ $\widehat{Y}$. This choice is also relevant because it serves as an upper bound of $I(\widetilde{Y}; Z_{\text{item}}, Z_{\text{user}})$. Notice that $\widetilde{Y}$ is a function of $\widehat{Y}$. Reducing $I(\widehat{Y}; Z_{\text{item}}, Z_{\text{user}})$ yields the minimization of the interested quantity $I(\widetilde{Y}; Z_{\text{item}}, Z_{\text{user}})$. For implementation of $I(\widehat{Y}; Z_{\text{item}}, Z_{\text{user}})$, we may employ the variational optimization technique in (Zhang et al., 2018; Cho et al., 2020b) to translate it into a function optimization which can also be parameterized. Other choices can also be dealt with properly relying upon the associated techniques in (Jiang et al., 2020; Mary et al., 2019).

## A.4 SYNTHETIC DATASET EXPERIMENTS

We consider a setting in which $(r, n, m) = (20, 600, 400)$. The synthetic data generated under the setting is randomly split into two subsets: 90% train set and 10% test set. Since $M_{ij} \in \{+1, -1\}$, we set the threshold $\tau = 0$, i.e., $\widetilde{Y} = \mathbf{1}\{\widehat{Y} \geq 0\}$. We train a matrix factorization (MF) based recommender system with the same rank as that of the dataset, i.e., $L \in \mathbb{R}^{600 \times 20}$ and $R \in \mathbb{R}^{20 \times 400}$. We set hyperparameters $(\delta, h) = (0.01, 0.01)$ and $\lambda = 0.99$ for KDE-based algorithm implementation. We use Adam optimizer for 1,000 iterations using full gradient. The learning rate is set to 1e-3 and $(\beta_1, \beta_2) = (0.9, 0.999)$. The additional experimental results in a variety of scenarios are listed from Table 4 to 8. We also visualize how the predicted preference rate of item groups for every user group $\Pr(\widetilde{Y} = 1 | Z_{\text{user}}, Z_{\text{item}})$ changes under our framework. See Fig. 3 and 4.

Table 4: Prediction error (RMSE) and fairness performances on the synthetic dataset. The dataset preserves observation bias $(q_0, q_1) = (0.2, 0.02)$ while exhibiting no population imbalance $(p_0, p_1) = (0.4, 0.4)$. The boldface indicates the best result and the underline denotes the second best. Each baseline, say VAL-based approach, enjoys the best fairness performance only for the measure focused therein, VAL, while it does not work well under the other fairness measures. On the other hand, our algorithm offers great fairness performances for all the measures, except for VAL, which our framework does not target.

| Measure | RMSE | DEE | VAL | UGF | CVS |
|---------|------|-----|-----|-----|-----|
| Unfair | $0.8423 \pm 0.0086$ | $0.1002 \pm 0.0264$ | $0.2992 \pm 0.0142$ | $0.0160 \pm 0.0031$ | $0.0066 \pm 0.0034$ |
| Ours (DEE) | $0.8611 \pm 0.0116$ | $\mathbf{0.0022 \pm 0.0003}$ | $0.3273 \pm 0.0043$ | $\underline{0.0006 \pm 0.0002}$ | $\underline{0.0003 \pm 0.0001}$ |
| Ours (DER) | $0.8605 \pm 0.0069$ | $0.0414 \pm 0.0059$ | $0.3130 \pm 0.0103$ | $0.0207 \pm 0.0030$ | $0.0001 \pm 0.0001$ |
| VAL-based | $0.8460 \pm 0.0081$ | $0.0829 \pm 0.0035$ | $\mathbf{0.0003 \pm 1.25e\text{-}5}$ | $0.0138 \pm 0.0019$ | $0.0066 \pm 0.0014$ |
| UGF-based | $0.8546 \pm 0.0050$ | $0.1137 \pm 0.0149$ | $0.3163 \pm 0.0036$ | $\mathbf{0.0004 \pm 0.0001}$ | $0.0195 \pm 0.0031$ |
| CVS-based | $0.8622 \pm 0.0072$ | $0.1189 \pm 0.0222$ | $0.3228 \pm 0.0076$ | $0.0220 \pm 0.0035$ | $\mathbf{0.0001 \pm 4.76e\text{-}5}$ |

Table 5: Prediction error (RMSE) and fairness performances on the synthetic dataset. The dataset preserves observation bias $(q_0, q_1) = (0.2, 0.03)$ while exhibiting no population imbalance $(p_0, p_1) = (0.4, 0.4)$.

| Measure | RMSE | DEE | VAL | UGF | CVS |
|---------|------|-----|-----|-----|-----|
| Unfair | $0.8031 \pm 0.0127$ | $0.0783 \pm 0.0053$ | $0.2248 \pm 0.0068$ | $0.0161 \pm 0.0043$ | $0.0073 \pm 0.0040$ |
| Ours (DEE) | $0.8240 \pm 0.0074$ | $\mathbf{0.0045 \pm 0.0032}$ | $0.2427 \pm 0.0089$ | $\underline{0.0018 \pm 0.0018}$ | $\underline{0.0005 \pm 0.0004}$ |
| Ours (DER) | $0.8287 \pm 0.0080$ | $0.0507 \pm 0.0086$ | $0.2495 \pm 0.0063$ | $0.0253 \pm 0.0043$ | $0.0001 \pm 0.0001$ |
| VAL-based | $0.8155 \pm 0.0100$ | $0.0779 \pm 0.0042$ | $\mathbf{0.0003 \pm 9.26e\text{-}6}$ | $0.0087 \pm 0.0014$ | $0.0082 \pm 0.0013$ |
| UGF-based | $0.8150 \pm 0.0111$ | $0.0910 \pm 0.0109$ | $0.2338 \pm 0.0092$ | $\mathbf{0.0008 \pm 0.0003}$ | $0.0211 \pm 0.0021$ |
| CVS-based | $0.8151 \pm 0.0115$ | $0.0791 \pm 0.0110$ | $0.2374 \pm 0.0092$ | $0.0234 \pm 0.0014$ | $\mathbf{0.0002 \pm 0.0001}$ |

Table 6: Prediction error (RMSE) and fairness performances on the synthetic dataset. The dataset preserves population imbalance $(p_0, p_1) = (0.4, 0.1)$ while exhibiting no observation bias $(q_0, q_1) = (0.2, 0.2)$.

| Measure | RMSE | DEE | VAL | UGF | CVS |
|---|---|---|---|---|---|
| Unfair | $0.0837 \pm 0.0149$ | $0.5859 \pm 0.0011$ | $0.0727 \pm 0.0009$ | $0.0193 \pm 0.0004$ | $0.0018 \pm 0.0005$ |
| Ours (DEE) | $0.6821 \pm 0.0025$ | $\mathbf{0.0123 \pm 0.0004}$ | $0.1865 \pm 0.0096$ | $\underline{0.0004 \pm 0.0002}$ | $0.0003 \pm 0.0002$ |
| Ours (DER) | $0.6761 \pm 0.0039$ | $0.0507 \pm 0.0052$ | $0.1885 \pm 0.0100$ | $0.0254 \pm 0.0026$ | $0.0004 \pm 0.0002$ |
| VAL-based | $0.3436 \pm 0.0110$ | $0.5648 \pm 0.0022$ | $\mathbf{0.0002 \pm 9.40e\text{-}6}$ | $0.0182 \pm 0.0007$ | $0.0018 \pm 0.0006$ |
| UGF-based | $0.5640 \pm 0.2033$ | $0.4495 \pm 0.1660$ | $0.0935 \pm 0.0527$ | $\mathbf{0.0001 \pm 3.62e\text{-}5}$ | $0.0047 \pm 0.0024$ |
| CVS-based | $0.1277 \pm 0.0107$ | $0.5856 \pm 0.0015$ | $0.0690 \pm 0.0005$ | $0.0188 \pm 0.0006$ | $\mathbf{0.0002 \pm 0.0001}$ |

Table 7: Prediction error (RMSE) and fairness performances on the synthetic dataset. The dataset preserves population imbalance $(p_0, p_1) = (0.4, 0.2)$ while exhibiting no observation bias $(q_0, q_1) = (0.2, 0.2)$.

| Measure | RMSE | DEE | VAL | UGF | CVS |
|---|---|---|---|---|---|
| Unfair | $0.0600 \pm 0.0102$ | $0.3789 \pm 0.0007$ | $0.0697 \pm 0.0004$ | $0.0217 \pm 0.0005$ | $0.0007 \pm 0.0003$ |
| Ours (DEE) | $0.6125 \pm 0.0038$ | $\mathbf{0.0115 \pm 0.0009}$ | $0.0938 \pm 0.0029$ | $\underline{0.0005 \pm 0.0001}$ | $0.0003 \pm 0.0002$ |
| Ours (DER) | $0.6187 \pm 0.0067$ | $0.0622 \pm 0.0015$ | $0.0894 \pm 0.0016$ | $0.0311 \pm 0.0008$ | $0.0006 \pm 0.0002$ |
| VAL-based | $0.3451 \pm 0.0109$ | $0.3684 \pm 0.0022$ | $\mathbf{0.0002 \pm 3.03e\text{-}6}$ | $0.0207 \pm 0.0011$ | $0.0033 \pm 0.0011$ |
| UGF-based | $0.4221 \pm 0.0130$ | $0.3713 \pm 0.0041$ | $0.0592 \pm 0.0020$ | $\mathbf{0.0002 \pm 2.38e\text{-}5}$ | $0.0058 \pm 0.0011$ |
| CVS-based | $0.1046 \pm 0.0059$ | $0.3790 \pm 0.0007$ | $0.0672 \pm 0.0012$ | $0.0219 \pm 0.0004$ | $\mathbf{0.0001 \pm 4.43e\text{-}5}$ |

Table 8: Prediction error (RMSE) and fairness performances on the synthetic dataset. The dataset preserves population imbalance $(p_0, p_1) = (0.4, 0.3)$ while exhibiting no observation bias $(q_0, q_1) = (0.2, 0.2)$.

| Measure | RMSE | DEE | VAL | UGF | CVS |
|---|---|---|---|---|---|
| Unfair | $0.0582 \pm 0.0098$ | $0.1952 \pm 0.0002$ | $0.0678 \pm 0.0006$ | $0.0233 \pm 0.0002$ | $0.0022 \pm 0.0005$ |
| Ours (DEE) | $0.4525 \pm 0.0058$ | $\mathbf{0.0049 \pm 0.0028}$ | $0.0767 \pm 0.0023$ | $\underline{0.0004 \pm 0.0003}$ | $0.0002 \pm 0.0001$ |
| Ours (DER) | $0.4946 \pm 0.0058$ | $0.0609 \pm 0.0049$ | $0.0827 \pm 0.0021$ | $0.0305 \pm 0.0025$ | $0.0003 \pm 0.0002$ |
| VAL-based | $0.3460 \pm 0.0046$ | $0.1863 \pm 0.0018$ | $\mathbf{0.0002 \pm 6.32e\text{-}6}$ | $0.0246 \pm 0.0010$ | $0.0030 \pm 0.0012$ |
| UGF-based | $0.4031 \pm 0.0095$ | $0.1830 \pm 0.0040$ | $0.0683 \pm 0.0024$ | $\mathbf{3.84e\text{-}5 \pm 2.62e\text{-}5}$ | $0.0039 \pm 0.0006$ |
| CVS-based | $0.1077 \pm 0.0178$ | $0.1927 \pm 0.0014$ | $0.0660 \pm 0.0008$ | $0.0228 \pm 0.0011$ | $\mathbf{0.0002 \pm 2.30e\text{-}5}$ |

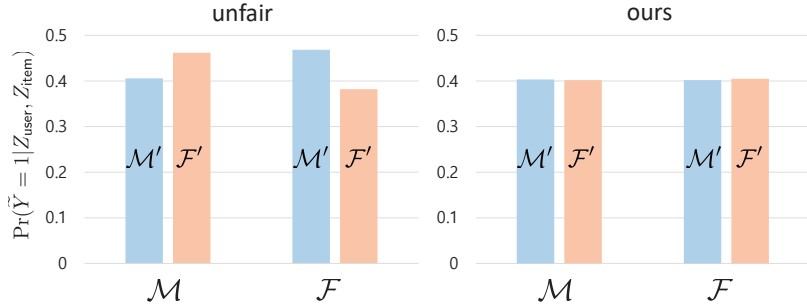

Figure 3: Predicted preference rate of item groups for every user group $\Pr(\tilde{Y} = 1 | Z_{\text{user}}, Z_{\text{item}})$ on the synthetic dataset. The dataset preserves observation bias $(q_0, q_1) = (0.2, 0.01)$ while exhibiting no population imbalance $(p_0, p_1) = (0.4, 0.4)$.

### A.5 REAL DATASETS EXPERIMENTS

- *MovieLens 1M*: The associated task is to predict the movie rating on a 5-star scale. This dataset contains 6,040 users, 3,900 movies, and 1,000,209 ratings, i.e., rating matrix is 4.26%

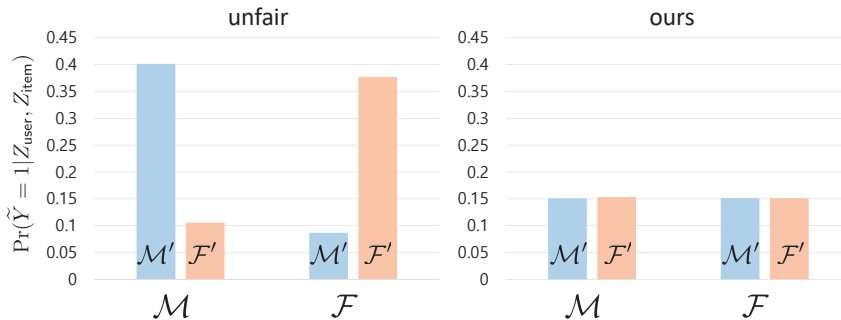

Figure 4: Predicted preference rate of item groups for every user group $\Pr(\tilde{Y} = 1|Z_{\text{user}}, Z_{\text{item}})$ on the synthetic dataset. The dataset preserves population imbalance $(p_0, p_1) = (0.4, 0.1)$ while exhibiting no observation bias $(q_0, q_1) = (0.2, 0.2)$.

full.[1] We divide user and item groups based on gender and genre, respectively. Action, crime, filme-noir, war are selected as male-preferred genre, whereas children, fantasy, musical, romance are selected as female-preferred genre. We can select male-preferred and female-preferred genres in a variety of ways based on ratings and observations. For various scenarios, the experimental results with similar trends are obtained, so we report the results for one representative scenario. If we assume that the real dataset is generated from the same model as the synthetic dataset, we can estimate the following probabilities. We empirically estimate the interested probabilities w.r.t. population imbalance as: $\widehat{p}_{\mathcal{MM'}} = 0.627$, $\widehat{p}_{\mathcal{MF'}} = 0.517$, $\widehat{p}_{\mathcal{FM'}} = 0.622$, and $\widehat{p}_{\mathcal{FF'}} = 0.595$. Similarly we obtain the estimates for the other probabilities w.r.t. observation bias: $\widehat{q}_{\mathcal{MM'}} = 0.053$, $\widehat{q}_{\mathcal{MF'}} = 0.037$, $\widehat{q}_{\mathcal{FM'}} = 0.037$, and $\widehat{q}_{\mathcal{FF'}} = 0.046$.

- *Last FM 360K*: The associated task is to predict whether the user likes the artist or not. This dataset contains 359,347 users, 294,015 artists, and 17,559,530 play counts, i.e., rating matrix is 0.02% full.[2] The data for play counts is converted to binary rating: $+1$ if counts $>$ average, otherwise $-1$. We divide user and item groups based on gender and genre, respectively. Since this dataset only contains gender information, we use Last.fm API[3] to collect the genre of corresponding artist's music; the tag was associated with 5,706 artists. We also randomly select 5000 male and 5000 female users. Among 10 genres, we choose hip-hop and musical for male and female preferred genres, respectively. The final rating matrix of 10,000 users and 5,706 artists is 0.55% full. From the real data, we obtain empirical estimates for the interested probabilities w.r.t. population imbalance: $\widehat{p}_{\mathcal{MM'}} = 0.548$, $\widehat{p}_{\mathcal{MF'}} = 0.421$, $\widehat{p}_{\mathcal{FM'}} = 0.438$ and $\widehat{p}_{\mathcal{FF'}} = 0.529$. Similarly we obtain the estimates for the other probabilities w.r.t. observation bias: $\widehat{q}_{\mathcal{MM'}} = 0.0054$, $\widehat{q}_{\mathcal{MF'}} = 0.0011$, $\widehat{q}_{\mathcal{FM'}} = 0.0036$ and $\widehat{q}_{\mathcal{FF'}} = 0.0038$.

We randomly split the real datasets into 90% train set and 10% test set. In case of MovieLens data, the rating is five-star based, so we set the threshold $\tau = 3$, i.e., $\widetilde{Y} = \mathbf{1}\{\widehat{Y} \geq 3\}$. On the other hand, for LastFM dataset, we set $\tau = 0$ as $M_{ij} \in \{+1, -1\}$. We run experiments employing both matrix factorization (MF) based and autoencoder (AE) based techniques. We set the rank as 512 for MF-based algorithm as was found by hyperparameter search. The structure of the employed autoencoder (Sedhain et al., 2015) is as follows: (i) encoder has two linear layers: 512 nodes with ReLU actiavation and 512 nodes with dropout layer (rate $= 0.7$) and ReLU activation; (ii) decoder has one layer with 512 nodes. For MovieLens 1M data (five-star ratings), we apply the clipping to the decoder output to fit into the range. For LastFM 360K data (binary rating: $+1$ and $-1$), we apply tanh activation. Hyperparameters for KDE-based algorithm are set to $(\delta, h) = (0.01, 0.01)$ and $\lambda = 0.9$. We use Adam optimizer for 1,000 iterations using full gradient, and the learning rate is

---

[1]http://www.movielens.org/

[2]http://ocelma.net/MusicRecommendationDataset/lastfm-360K.html

[3]http://www.last.fm/api

set to 1e-3. Since the main paper contains mostly MF-based experiments, here we only present the performances of *autoencoder* based algorithm on both real datasets.

Table 9: Prediction error (RMSE) and fairness performances of the *autoencoder* based algorithm on MovieLens 1M dataset. We observe the same performance trends as those in Table 4.

| Measure | RMSE | DEE | VAL | UGF | CVS |
|---|---|---|---|---|---|
| Unfair | $0.8369 \pm 0.0012$ | $0.2477 \pm 0.0175$ | $0.3412 \pm 0.0031$ | $0.0419 \pm 0.0042$ | $0.1158 \pm 0.0025$ |
| Ours (DEE) | $0.8437 \pm 0.0042$ | $\mathbf{0.0120 \pm 0.0028}$ | $0.3338 \pm 0.0037$ | $\underline{0.0039 \pm 0.0022}$ | $0.0042 \pm 0.0010$ |
| Ours (DER) | $0.8411 \pm 0.0027$ | $0.0285 \pm 0.0084$ | $0.3395 \pm 0.0048$ | $0.0144 \pm 0.0046$ | $0.0061 \pm 0.0023$ |
| VAL-based | $0.8433 \pm 0.0022$ | $0.2138 \pm 0.0363$ | $\mathbf{0.2128 \pm 0.0143}$ | $0.0299 \pm 0.0172$ | $0.0918 \pm 0.0070$ |
| UGF-based | $0.8491 \pm 0.0056$ | $0.1934 \pm 0.0109$ | $0.3391 \pm 0.0040$ | $\mathbf{0.0011 \pm 0.0006}$ | $0.0982 \pm 0.0055$ |
| CVS-based | $0.8495 \pm 0.0050$ | $0.0808 \pm 0.0225$ | $0.3424 \pm 0.0069$ | $0.0343 \pm 0.0105$ | $\mathbf{0.0023 \pm 0.0010}$ |

Table 10: Prediction error (RMSE) and fairness performances of the *autoencoder* based algorithm on Last FM 360K dataset.

| Measure | RMSE | DEE | VAL | UGF | CVS |
|---|---|---|---|---|---|
| Unfair | $0.6534 \pm 0.0024$ | $0.1003 \pm 0.0172$ | $0.2253 \pm 0.0031$ | $0.0501 \pm 0.0086$ | $0.0006 \pm 0.0002$ |
| Ours (DEE) | $0.6649 \pm 0.0212$ | $\mathbf{0.0024 \pm 0.0005}$ | $0.2213 \pm 0.0133$ | $\underline{0.0012 \pm 0.0008}$ | $0.0007 \pm 0.0006$ |
| Ours (DER) | $0.6501 \pm 0.0004$ | $0.0878 \pm 0.0015$ | $0.2204 \pm 0.0004$ | $0.0439 \pm 0.0008$ | $0.0008 \pm 0.0001$ |
| VAL-based | $0.6828 \pm 0.0202$ | $0.0571 \pm 0.0054$ | $\mathbf{0.1915 \pm 0.0038}$ | $0.0288 \pm 0.0029$ | $0.0063 \pm 0.0029$ |
| UGF-based | $0.6861 \pm 0.0310$ | $0.0485 \pm 0.0025$ | $0.2098 \pm 0.0143$ | $\mathbf{0.0001 \pm 5.00e\text{-}5}$ | $0.0021 \pm 0.0016$ |
| CVS-based | $0.6685 \pm 0.0079$ | $0.0960 \pm 0.0096$ | $0.2421 \pm 0.0105$ | $0.0480 \pm 0.0047$ | $\mathbf{0.0002 \pm 3.24e\text{-}5}$ |

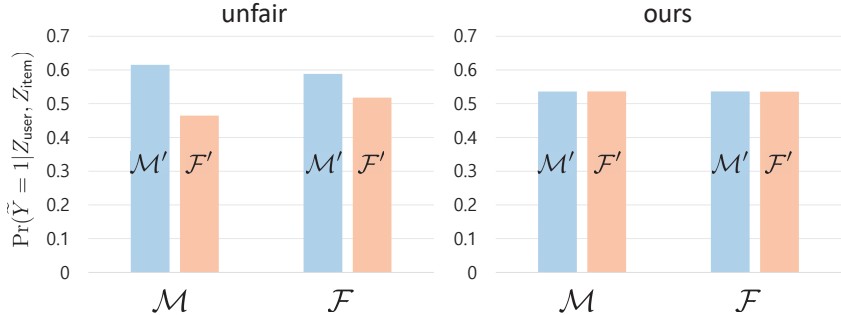

Figure 5: Predicted preference rate of item groups for every user group $\Pr(\tilde{Y} = 1 | Z_{\text{user}}, Z_{\text{item}})$ of the matrix factorization based algorithm on *MovieLens 1M dataset*.

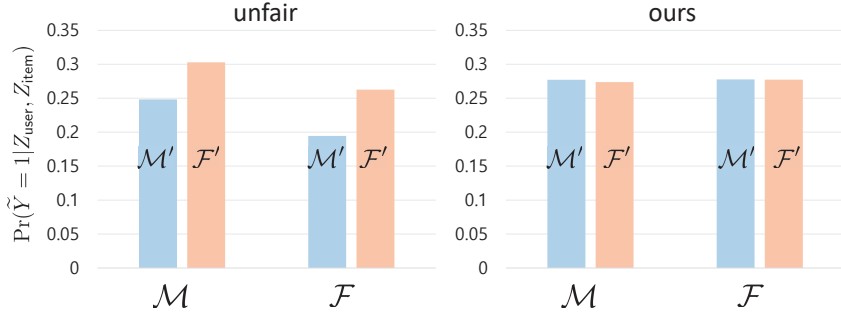

Figure 6: Predicted preference rate of item groups for every user group $\Pr(\tilde{Y} = 1 | Z_{\text{user}}, Z_{\text{item}})$ of the matrix factorization based algorithm on *Last FM 360K dataset*.

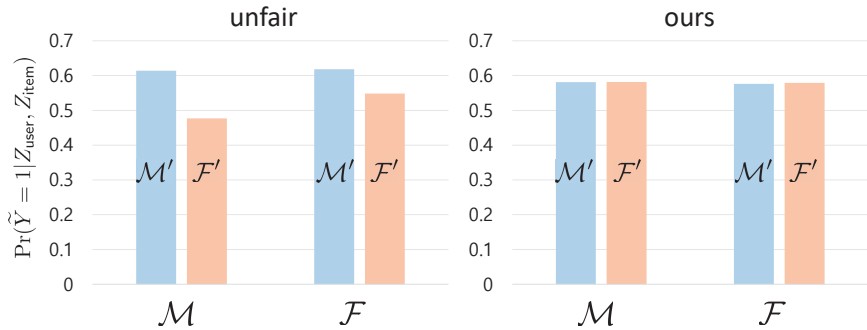

Figure 7: Predicted preference rate of item groups for every user group $\Pr(\tilde{Y} = 1 | Z_{\text{user}}, Z_{\text{item}})$ of the *autoencoder* based algorithm on *MovieLens 1M dataset*.

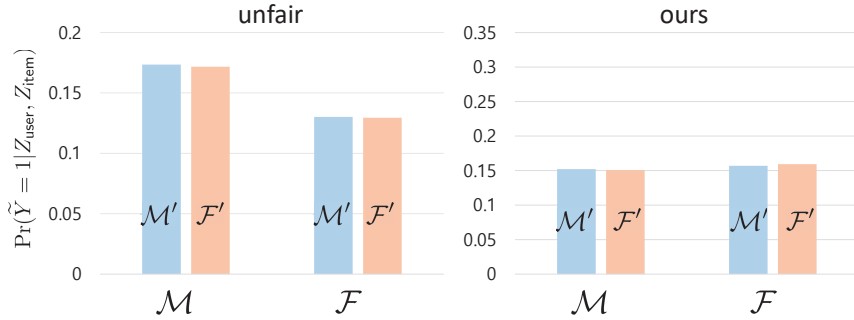

Figure 8: Predicted preference rate of item groups for every user group $\Pr(\tilde{Y} = 1 | Z_{\text{user}}, Z_{\text{item}})$ of the *autoencoder* based algorithm on *Last FM 360K dataset*.

### A.6 COMPLEXITY ANALYSIS

We do complexity analysis of ours in light of other baselines. For comparison, we consider the running time measured under Pytorch on Xeon Silver 4210R CPU and TITAN RTX GPU. Table 11 presents the running times of matrix factorization based algorithms on the synthetic and real datasets. While our approach provides better fairness performance w.r.t. DEE (as in the above tables), it comes at a cost of an increased complexity, around twice relative to the CVS-based algorithm.

Table 11: The running time (in seconds) of our algorithm and baselines on the synthetic and two real datasets: MovieLens 1M and LastFM 360K.

| Dataset | Synthetic | MovieLens 1M | LastFM 360K |
|---|---|---|---|
| Unfair | 2.15 | 6.72 | 16.27 |
| Ours (DEE) | 13.23 | 86.82 | 192.02 |
| VAL-based (Yao & Huang, 2017) | 5.83 | 201.08 | 477.14 |
| UGF-based (Li et al., 2021) | 7.05 | 60.14 | 136.69 |
| CVS-based (Kamishima & Akaho, 2017) | 7.16 | 47.89 | 104.42 |

### A.7 EXTENSION TO THE FAIRNESS NOTION: $\widetilde{Y} \perp Z_{\text{user}}, Z_{\text{item}} | Y$

Similar to DEE in equation 12, we quantify the notion via:

$$\sum_{y \in \{0,1\}} \sum_{z_1 \in \mathcal{Z}_{\text{user}}} \sum_{z_2 \in \mathcal{Z}_{\text{item}}} \left| \mathbb{P}(\widetilde{Y} = 1 | Y = y) - \mathbb{P}(\widetilde{Y} = 1 | Y = y, Z_{\text{item}} = z_2, Z_{\text{user}} = z_1) \right|, \quad (21)$$

for arbitrary alphabet sizes $|\mathcal{Z}_{\text{user}}|$ and $|\mathcal{Z}_{\text{item}}|$.

Using the KDE approach, similarly we can obtain:

$$\widehat{\mathbb{P}}(\widetilde{Y} = 1 | Y = y) = \int_{\tau}^{\infty} \widehat{f}_{\widehat{Y}|Y}(\hat{y}|y)d\hat{y} = \frac{1}{|I_y|} \sum_{(i,j) \in I_y} F_k\left(\frac{\tau - \widehat{M}_{ij}}{h}\right),$$

where $F_k(\hat{y}) := \int_{\hat{y}}^{\infty} f_k(t)dt$; $Y_{ij} := \mathbf{1}\{M_{ij} \geq \tau\}$; and $I_y := \{(i,j) : (i,j) \in \Omega, Y_{ij} = y\}$. We can then compute the gradien w.r.t. $w$ as:

$$\nabla_w \widehat{\mathbb{P}}(\widetilde{Y} = 1 | Y = y) = \frac{1}{|I_y|h} \sum_{(i,j) \in I_y} f_k\left(\frac{\tau - \widehat{M}_{ij}}{h}\right) \cdot \nabla_w \widehat{M}_{ij}. \tag{22}$$

We can then enjoy a family of gradient-based optimizers (Géron, 2017; Kingma & Ba, 2014a). We provide experimental results for the extension on MovieLens 1M (Harper & Konstan, 2015) real dataset. We run experiments employing both matrix factorization (MF) based and autoencoder (AE) based techniques. We demonstrate that the framework based on the extended notion can indeed mitigate such unfairness while exhibiting a minor degradation of recommendation accuracy. The results are listed in Table 12 and 13.

Table 12: Prediction error (RMSE) and fairness performances of the matrix factorization based algorithm on *MovieLens 1M dataset*. The boldface indicates the best result and the underline denotes the second best. The approach based on the extended fairness notion (conditioning on $Y$), enjoys the best fairness performance for the measure focused therein.

| Measure | RMSE | DEE | Conditioning on $Y$ (21) | VAL | UGF | CVS |
|---|---|---|---|---|---|---|
| Unfair | $0.8541 \pm 0.0033$ | $0.2447 \pm 0.0134$ | $0.3494 \pm 0.0071$ | $0.3227 \pm 0.0031$ | $0.0058 \pm 0.0042$ | $0.1291 \pm 0.0079$ |
| Ours (DEE) | $0.8641 \pm 0.0047$ | $\mathbf{0.0014 \pm 0.0008}$ | $0.3005 \pm 0.0048$ | $0.2941 \pm 0.0024$ | $\underline{0.0018 \pm 0.0016}$ | $\underline{0.0007 \pm 0.0004}$ |
| Conditioning on $Y$ | $0.8576 \pm 0.0011$ | $0.1451 \pm 0.0079$ | $\mathbf{0.0283 \pm 0.0057}$ | $0.3230 \pm 0.0026$ | $0.0052 \pm 0.0029$ | $0.0668 \pm 0.0051$ |
| VAL-based | $0.8529 \pm 0.0011$ | $0.3659 \pm 0.0033$ | $0.4679 \pm 0.0098$ | $\mathbf{0.0942 \pm 0.0016}$ | $0.0261 \pm 0.0020$ | $0.1388 \pm 0.0030$ |
| UGF-based | $0.8550 \pm 0.0015$ | $0.2492 \pm 0.0100$ | $0.3657 \pm 0.0084$ | $0.3285 \pm 0.0051$ | $\mathbf{0.0001 \pm 0.0001}$ | $0.1355 \pm 0.0038$ |
| CVS-based | $0.8549 \pm 0.0018$ | $0.0721 \pm 0.0069$ | $0.3260 \pm 0.0135$ | $0.3319 \pm 0.0046$ | $0.0065 \pm 0.0042$ | $\mathbf{0.0002 \pm 3.45e\text{-}5}$ |

Table 13: Prediction error (RMSE) and fairness performances of the *autoencoder* based algorithm on MovieLens 1M dataset.

| Measure | RMSE | DEE | Conditioning on $Y$ (21) | VAL | UGF | CVS |
|---|---|---|---|---|---|---|
| Unfair | $0.8369 \pm 0.0012$ | $0.2477 \pm 0.0175$ | $0.3557 \pm 0.0086$ | $0.3412 \pm 0.0031$ | $0.0419 \pm 0.0042$ | $0.1158 \pm 0.0025$ |
| Ours | $0.8437 \pm 0.0042$ | $\mathbf{0.0120 \pm 0.0028}$ | $0.2260 \pm 0.0203$ | $0.3338 \pm 0.0037$ | $\underline{0.0039 \pm 0.0022}$ | $0.0042 \pm 0.0010$ |
| Conditioning on $Y$ | $0.8467 \pm 0.0012$ | $0.1157 \pm 0.0203$ | $\mathbf{0.0505 \pm 0.0264}$ | $0.3395 \pm 0.0039$ | $0.0131 \pm 0.0139$ | $0.0551 \pm 0.0046$ |
| VAL-based | $0.8433 \pm 0.0022$ | $0.2138 \pm 0.0363$ | $0.3372 \pm 0.0117$ | $\mathbf{0.2128 \pm 0.0143}$ | $0.0299 \pm 0.0172$ | $0.0918 \pm 0.0070$ |
| UGF-based | $0.8491 \pm 0.0056$ | $0.1934 \pm 0.0109$ | $0.3285 \pm 0.0178$ | $0.3391 \pm 0.0040$ | $\mathbf{0.0011 \pm 0.0006}$ | $0.0982 \pm 0.0055$ |
| CVS-based | $0.8495 \pm 0.0050$ | $0.0808 \pm 0.0225$ | $0.2408 \pm 0.0165$ | $0.3424 \pm 0.0069$ | $0.0343 \pm 0.0105$ | $\mathbf{0.0023 \pm 0.0010}$ |

