# OpenReview forum: "Equal Experience in Recommender Systems"
_ICLR.cc/2022/Conference — ICLR 2022 Submitted_

### Official Review · Reviewer_TDXy · 2021-11-02

**Correctness:** 3
**Technical Novelty And Significance:** 2
**Empirical Novelty And Significance:** 2
**Recommendation:** 6
**Confidence:** 3

**Main Review:**

Strengths:
+ The paper is well written and easy to follow.
+ The related works is well described, especially the representative ones. Fair comparisons on these representative baselines are provided in the experiments.
+ It is interesting to utilize mutual information for fairness problems.

Weaknesses:
- I suggest to include DER to be a baseline.
- I am wondering if there are any difficulties when facing more than group of users and items.
- As shown in the experiments, the proposed method keeps receiving the worst RMSE performance. Also, from table 6, we can observe that when the proposed performs much better than  the baselines for fairness comparisons, it also performs much worse than others for recommendation comparisons. There is tradeoff between fairness and accuracy, which may need a more detailed analysis.

**Summary Of The Paper:**

This paper proposes a new notion “equal experience” to measure the fairness among groups in recommender systems, and provides an method to optimize this new notion based on matrix completion. Experiments demonstrate the effective of the proposed optimization framework.

**Summary Of The Review:**

- Since I am not very familiar with recent work on fairness, the proposed notion "equal experience" seems novel to me.
- The authors provide the optimization method for the "equal experience"  based on both traditional ML and deep learning techniques.
- The results prove that the proposed method can receive better "equal experience" optimization performance compared to other baselines.

---

> ### Author Response · Authors · 2021-11-23
> **Author's Response to Reviewer TDXy**
>
> We do appreciate your very constructive suggestions. Below we provide detailed responses as to how we will implement them.
>
> [4-1] *(Suggestion: including DER-based approach as a baseline)*: As per your great suggestion, we now included a DER-based approach as a baseline in the revision. See Section 4 for changes, marked in blue.
>
> [4-2] *(Extension to multiple user/item groups)*: Our framework encompasses such a generalized setting by allowing for an arbitrary alphabet size of sensitive attributes. The implementation for the multiple groups is similar to that for the binary group size. For instance, for different races like White, Black, Asian, and Hispanic, we can set $\|\mathcal{Z}_{\sf user}\|=4$ in equation (12) and then apply our algorithm accordingly.
>
> [4-3] *(Tradeoff between fairness and prediction accuracy)*: The proposed fairness notion ensures four types of independence while the baselines ensure only one type. Hence, more constraints considered in our framework yield worse RMSE performance compared to the baselines. However, ours is competitive to the baselines except for a few cases with extreme “population imbalance”. The RMSE gap between ours and the baselines is less than 1.5% in all real dataset experiments.

---

### Official Review · Reviewer_dddS · 2021-11-02

**Correctness:** 4
**Technical Novelty And Significance:** 2
**Empirical Novelty And Significance:** 2
**Recommendation:** 3
**Confidence:** 3

**Main Review:**

I think that the paper solves an interesting problem and generalizes previous work. The main concern here seems to be with the weight of the contribution. Specifically, the new notion of fairness is not very complicated and seems to follow from previous work. Further, the optimization algorithms also seems to follow from previous work and are not novel. If there were significant technical issues it does not seem to me that they were well-emphasized in the paper.

Have the authors thought about including any theoretical guarantees. For example, convergence of the optimization. Or a guarantee that the fairness notion would hold at least approximately.

Some minor representation issues:
1-Eq(2) on page 3, shouldn't \hat{M} belong to a constraint set such low rank.
2-Eq(4) on page 3, I think (M_{ij}-\hat{M}_{ij}) is missing a square.





**Summary Of The Paper:**

The paper is concerned with fairness in recommendations. Specifically, there are groups of users and groups of items. Previous work has modelled fairness as the constraint of all user groups having the same accuracy or as the prediction probability being independent of the item group or the user group. In this work, the notion is generalized so that the prediction is independent of both the item and user group. Optimization algorithms are shown to solve this problem along with experimental results.

**Summary Of The Review:**

While the paper introduces a new meaningful notion of fairness. The notion closely follows the previous work. The optimization methods used also follow the previous work. Overall, this makes the contribution of the paper very incremental.

---

> ### Author Response · Authors · 2021-11-23
> **Author's Response to Reviewer dddS**
>
> We would like to thank the reviewer for providing comments and pointing out unclear points.
>
> [3-1] (*Contribution of this paper*): The key motivation of the proposed fairness notion is to ensure all of the following four types of independence (See Section 3.1):
> \begin{equation}
> \widetilde{Y}\perp Z_{\sf item},\quad  \widetilde{Y}\perp Z_{\sf user}|Z_{\sf item},\quad  \widetilde{Y}\perp Z_{\sf user},\quad  \widetilde{Y}\perp Z_{\sf item}|Z_{\sf user}.
> \end{equation}
> Since the form of $I(\widetilde{Y}; Z_{\sf user}, Z_{\sf item})$ is similar to that of $I(\widetilde{Y}; Z_{\sf item})$, it looks like a simple extension. However, we believe it is not the case. The reason that we end up with the simple formula $I(\widetilde{Y}; Z_{\sf user}, Z_{\sf item})$ is actually due to the use of the chain rule of mutual information which summarizes all the complicated details.
>
> [3-2] (*Re. theoretical guarantees of the algorithm*):
>   Our main contributions lie in introducing the new novel fairness notion and the framework. The proposed framework can be gracefully merged with a variety of measure-based fairness algorithms (e.g., covariance, mutual information, Wasserstein distance). Theoretical guarantees for these widely-used algorithms have been provided in the prior fairness literature. So we did not focus on the analysis of an algorithm that we have chosen as a particular choice. Instead, we intended to deliver a clear motivation behind the proposed notion, as well as to demonstrate via extensive experiments that our framework indeed mitigates unfairness.
>
> [3-3] (*Minor representation issues*)
>
> equation 2: We did not assume the low rank constraint as in [Koren et al., 2009]. So we consider an optimization that contains only the Frobenius norm w.r.t. the ground truth and its estimate.
>
> equation 4: It is not a typo. We employ the same value unfairness (defined in equation 6 from [Yao & Huang, 2017]) which contains no square term therein.

---

### Official Review · Reviewer_85io · 2021-11-02

**Correctness:** 4
**Technical Novelty And Significance:** 2
**Empirical Novelty And Significance:** 2
**Recommendation:** 6
**Confidence:** 4

**Main Review:**

The presented study did not analyse the diversity of the recommended items in the experimental results, which the new proposed fairness notion aims to improve. Also, it would further strengthen the study if the authors could analyse who could benefit from the proposed method, e.g., females who enjoyed action or crime movies but did not have much historical data, or others?

Also, as this study focused on the problem of rating prediction in recommender systems, it would good to also test the effectiveness of the proposed method in the ranking setting.

**Summary Of The Paper:**

The presented study argued that a fair recommendation should be independent of both user and item. Therefore, the study introduced a new fairness notion, i.e., equal experience, and further incorporated this fairness notion as a regularisation term in the matrix completion framework to construct a fair recommender system. The proposed method was evaluated with three datasets (one synthetic and two real datasets).

**Summary Of The Review:**

The new fairness notion introduced by the presented study can provide certain new knowledge and insights on how to construct a fair recommender system, and the effectiveness of the proposed recommendation method seemed to be supported by the experimental results. However, the study can be further strengthened if more in-depth analysis on the recommendation results can be provided.

---

> ### Author Response · Authors · 2021-11-23
> **Author's Response to Reviewer 85io**
>
> We would like to thank the reviewer for the constructive feedback and detailed suggestions, which helped us improve the manuscript. We provide point-by-point replies below.
>
> [2-1] (*Analysis: diversity of the recommended items*): As per your great suggestion, we now conducted additional experimental results to analyze the diversity of the recommended items. In the revision, we also visualized how the predicted preference rate of item groups for every user group, i.e., $\Pr(\widetilde{Y}=1|Z_{\sf user}, Z_{\sf item})$, changes under our framework. In Appendices A.4 and A.5, we highlighted the changes marked in blue.
>
> [2-2] (*Re. testing in a ranking setting*): Many end-to-end recommender systems offer a recommendation list often via two processes: (i) candidate generation, and (ii) ranking. As an initial effort, we focus on fair candidate generation that we built collaborative filtering for in this work. However, we agree that testing under the ranking setting would certainly improve our paper. Unfortunately, we employed rating-based real datasets which do not provide ranking information of items. We thus leave testing as future work (See Section 5).

---

### Official Review · Reviewer_zEp7 · 2021-11-04

**Correctness:** 3
**Technical Novelty And Significance:** 3
**Empirical Novelty And Significance:** 3
**Recommendation:** 5
**Confidence:** 4

**Main Review:**

Strengths: The result due to chain rule is pretty interesting and neat in that joint independence of preference prediction user-item groups ensures multiple independence notions together. The authors do a good job of giving examples of the definition in terms of all of the independence notions.

Empirically, the method proposed in the paper gives good fairness performance on all fairness metrics compared (except one), while the specific papers only specialize in reducing one form of unfairness at a time.

Weaknesses:
The normative goal of the fairness notion is not well-motivated in the paper. From what I understand, making the preference prediction independent of both the item group and the user group means that different base rates of preference for item groups among different user groups should not mean different rates of recommendation. The authors give an example of science vs literature preference for male and female users, where according to this measure, science and literature would be recommended at the rate at which they appear in the item set (because of the independence between Y~ and Z_item). In my opinion, this notion needs to be further motivated and included with a proper discussion about the scope of what item and user groups could mean. For example, if there are more literature courses to be recommended than science courses, they might not have the same capacity or preference among the general user group as science courses. While it may make sense to recommend science and literature equally to male and female users, wouldn't recommending science and literature at proportional rates only make sense when they are equally preferred in the general population?

In the spirit of the above question, I would like the authors to elaborate on their criticism of fairness notions based on the difference in recommendation accuracies (Paragraph 2 of Introduction). However, later in the future work (Sec 5), the authors seem to advocate for Equalized odds fairness notion which is an accuracy-based notion.

The setup uses Y>threshold as a preference prediction isn't the most used form of recommendation, while something like the relative ranking/ordering of the items is. The paper does not talk about how the defined fairness metrics would apply if predictions were used in terms of ranking the different items for each user/user type.

In the experiments section, would it also make sense to use an importance weighting approach (e.g. [1]) to tackle the selection bias created by the data generation process? [1] Schnabel, Tobias, et al. "Recommendations as treatments: Debiasing learning and evaluation." ICML, 2016.

Minor:
- In paragraph 2 of the Introduction, the authors say: “female students exhibit low ratings on math and science subjects due to …. sampling bias”. It is not clear if the term "ratings" means the predictions or scores observed in a dataset. However, the other factor of societal/cultural differences mentioned by the authors does explain the observation well. Perhaps, the authors meant the lack of ratings instead of lower ratings.
- In related work, some literature from debiasing word embeddings could be relevant to include because those works also include an independence-based notion between embeddings of occupations and gender-specific words. In the current work, since matrix completion is used as the collaborative filtering method, it also corresponds to user and item embeddings and the notions of orthogonality and independence might be relevant to mention (or describe).

**Summary Of The Paper:**

The paper defines a fairness notion for a recommender system that is based on the independence of preference predictions to the user group and the item groups. The paper defines a mutual information-based mathematical expression to measure unfairness, called equal experience metric, and then optimize it along with a matrix factorization-based collaborative filtering.

**Summary Of The Review:**

The paper tackles an important problem of fairness in recommender systems, and it defines a fairness notion that contains both the user groups and items groups. The metric defined is pretty straightforward to understand. The kernel-based probability density estimation to compute the difference between conditional and marginal probabilities is not a very standard method but it seems to work for the purposes of the experiment. Overall, the fairness notions are not very strongly motivated (as highlighted above), and the paper is missing the guidance around what groupings of items and users are meaningful for such a recommendation task.

---

> ### Author Response · Authors · 2021-11-23
> **Author's Response to Reviewer zEp7**
>
>  We would like to express our sincere gratitude for the constructive feedback and detailed suggestions, which helped us improve the manuscript. We provide point-by-point replies below.
>
> [1-1] (*The motivation of the proposed fairness notion*): As the reviewer commented, in general the population exhibits a higher preference for a specific item group. Nevertheless, if the recommended items are selected as per the *overall* preference, the biased preference for a specific item group will further increase, and the exposure to the unpreferred item group will gradually decrease. To prevent this, we regularize the proportional recommendation rate for each item group so as to yield almost the same rate while minimizing recommendation accuracy degradation. As per your great suggestion, we now clarified our motivation in the revision. See Section 1 for changes, marked in blue.
>
> [1-2] (*Extension to $\widetilde{Y} \perp Z_{\sf user}, Z_{\sf item}|Y$*): Because of the reason that we mentioned in [1-1], we believe that ensuring the proposed fairness notion is important. On the other hand, one may want to yield the same accurate rate. This is why we also advocated the Equalized Odds fairness notion (accuracy-based one). So we wanted to emphasize that our notion and framework can be easily extensible to such an accuracy-based notion (See Appendix A.7). One can easily choose the notion and framework depending on one’s purpose.
>
> [1-3] (*Extension to a ranking setup*): Many end-to-end recommender systems offer a recommendation list via two processes: (i) candidate generation, and (ii) ranking. In this work, we focus on the first candidate generation for which we built collaborative filtering. But our proposed notion can also be applicable in generating an end ranked list. The idea behind the end-ranked list generation is to define $\widetilde{Y}$ as an indicator function which returns 1 when the item of interest belongs to, say top-k item set (0 otherwise). In this case, the same notion $\widetilde{Y}\perp (Z_{\sf user}, Z_{\sf item})$ serves a proper role. The ranking setup is out of the scope of this work; hence, we do not include detailed implementation of the ranking-based notion (See Section 5).
>
> [1-4] (*Re. importance weighting approach*): Mitigating selection bias (that we call observation bias in the paper) would be one of the methods that one can employ as a baseline. Here we provide experimental results on the synthetic dataset for the importance weighting approach in [1]. We set $(p_0,p_1)=(0.4,0.1)$ and $(q_0,q_1)=(0.2,0.2)$ (See Table 6).
>
> \begin{matrix}
> \text{Measure} & \text{RMSE} & {\sf DEE} & {\sf VAL}  & {\sf UGF} & {\sf CVS} \\\\
> \hline
> \text{Unfair}   & 0.0837 \pm 0.0149 & 0.5859 \pm 0.0011 & 0.0727 \pm 0.0009 & 0.0193 \pm 0.0004 & 0.0018 \pm 0.0005 \\\\
>     \text{Ours}  & 0.6821 \pm 0.0025 & \textbf{0.0123} \pm \textbf{0.0004} & 0.1865 \pm 0.0096 & \underline{0.0004 \pm 0.0002}& \underline{0.0003 \pm 0.0002}\\\\
>     {\sf VAL}\text{-based} & 0.3436 \pm 0.0110 & 0.5648 \pm 0.0022 & \textbf{0.0002} \pm \textbf{9.40e-6} & 0.0182 \pm 0.0007 & 0.0018 \pm 0.0006\\\\
>     {\sf UGF}\text{-based}   & 0.5640 \pm 0.2033 & 0.4495 \pm 0.1660 & 0.0935 \pm 0.0527 & \textbf{0.0001} \pm \textbf{3.62e-5} & 0.0047 \pm 0.0024 \\\\
>     {\sf CVS}\text{-based}   & 0.1277 \pm 0.0107 & 0.5856 \pm 0.0015 & 0.0690 \pm 0.0005 & 0.0188 \pm 0.0006 & \textbf{0.0002} \pm \textbf{0.0001} \\\\
> \textcolor{RedOrange}{\text{IWA}}   & \textcolor{RedOrange}{0.1085 \pm 0.0150} &   \textcolor{RedOrange}{0.5847 \pm 0.0013} &\textcolor{RedOrange}{0.0716 \pm 0.0007} & \textcolor{RedOrange}{0.0190 \pm 0.0004}      &\textcolor{RedOrange}{0.0017 \pm 0.0005}
> \end{matrix}
>
> We employ observation probabilities ($q_0$, $q_1$) for weighting in the approach. Since the approach (IWA) only considers selection bias without population imbalance, we observe that the approach does not mitigate unfairness w.r.t. DEE. In this paper, we consider fairness-measure-based algorithms as baselines. So we included this approach only in related works instead of adding it as a baseline in the revision. See “Related works” for changes, marked in blue.
> [1] Schnabel, Tobias, et al. "Recommendations as treatments: Debiasing learning and evaluation." ICML, 2016.
>
> [1-5] (*Clarity: re. sampling biases*): We do appreciate your constructive feedback. There are two meanings: (1) lower ratings, and (2) lack of ratings. For clarity, we fixed the sentence as follows. “For instance, in the subject recommendation, the fairness notion may not serve properly, as long as female students exhibit *low ratings (and/or lack of ratings)* on math and science subjects due to *societal/cultural influences (and/or sampling biases)*.”

---

### Decision · Program_Chairs · 2022-01-20

**Decision:**

Reject

**Comment:**

This paper introduces a new (un)fairness metric for recommender systems based on mutual information and then develop an algorithm to account for this metric in matrix factorization-based collaborative filtering. The reviewers all agree that the proposed metric and algorithm are sound at a technical level, however, they have concerns regarding the motivation of the introduced metric as well as the experimental evaluation. The rebuttal by the authors did not persuade the reviewers to reconsider their original assessment and they still argued that their concerns remained. In the final recommendation, the simplicity of the metric was not seeing as a weakness of the work.